# Characterization of Flavor Compounds in Chinese Indigenous Sheep Breeds Using Gas Chromatography–Ion Mobility Spectrometry and Chemometrics

**DOI:** 10.3390/foods13172647

**Published:** 2024-08-23

**Authors:** Fang Wang, Hongbo Wang, Zeyi Liang, Jing Liu, Chen Yang, Huan Zhai, Anle Chen, Zengkui Lu, Yaqin Gao, Xuezhi Ding, Jianbin Liu

**Affiliations:** 1Key Laboratory of Veterinary Pharmaceutical Development, Ministry of Agriculture and Rural Affairs, Lanzhou Institute of Animal Husbandry and Pharmaceutical Science, Chinese Academy of Agricultural Sciences, Lanzhou 730050, China; wf13359317003@163.com (F.W.); liangzeyi1@163.com (Z.L.); 82101215421@caas.cn (J.L.); yangchen19960905@163.com (C.Y.); zhaihhuan@163.com (H.Z.); cal_ycsa@163.com (A.C.); 2Laboratory of Quality & Safety Risk Assessment for Livestock Products of Ministry of Agriculture, Lanzhou Institute of Animal Husbandry and Pharmaceutical Science, Chinese Academy of Agricultural Sciences, Lanzhou 730050, China13609312788@163.com (Y.G.); 3Key Laboratory of Animal Genetics and Breeding on Tibetan Plateau, Ministry of Agriculture and Rural Affairs, Lanzhou Institute of Animal Husbandry and Pharmaceutical Science, Chinese Academy of Agricultural Sciences, Lanzhou 730050, China; luzengkui@caas.cn; 4Sheep Breeding Engineering Technology Research Center of Chinese Academy of Agricultural Sciences, Lanzhou 730050, China

**Keywords:** Chinese indigenous sheep, volatile compounds, GC-IMS, principal component analysis, chemometric

## Abstract

This study analyzed the flavor compounds in the meat of four indigenous breeds of Chinese sheep through the use of gas chromatography–ion mobility spectrometry (GC-IMS). GC-IMS provided information on the characteristics and strength of 71 volatile flavor compounds (monomers and dimers), with aldehydes, alcohols and ketones being the most abundant in all types of sheep meat. The compounds with higher intensity peaks in the sheep meat were aldehydes (n-nonanal, octanal, heptanal, 3-methylbutanal, and hexanal), alcohols (1-octen-3-ol, hexanol, and pentanol), ketones (3-hydroxy-2-butanon, 2-butanone, and 2-propanone), esters (methyl benzoate), and thiazole (trimethylthiazole). The volatile flavor components in the meat of the different breeds of sheep obtained via GC-IMS were further differentiated using principal component analysis. In addition, orthogonal partial least squares discriminant analysis (OPLS-DA) and variable importance on projection (VIP) were used to determine the characteristic flavor compounds in the meats of different breeds of sheep, and 21 differentially volatile components were screened out based on having a VIP above 1. These results indicate that GC-IMS combined with multivariate analysis is a convenient and powerful method for characterizing and discriminating sheep meat.

## 1. Introduction

Currently, with the largest quantity of sheep sales and highest mutton output worldwide, China boasts a rich variety of sheep breeds and genetic resources [1]. Tibetan sheep, Ujumqin sheep, Tan sheep, and Hu sheep, as excellent indigenous breeds of sheep, have been documented in the National Breed List of Livestock and Poultry Genetic Resources. These breeds have gained popularity in both the Chinese and global markets for their efficient feed conversion rates and economic advantages. It is important to note that meat quality and economic worth can vary significantly between various breeds of sheep [2]. Flavor is a crucial quality attribute of meat and the most crucial factor influencing consumers’ meat purchasing habits [3].

Compared with taste and texture, it is better to use volatile flavor as an index to distinguish the quality features of meat products [4]. The abundant flavor components in meat products are generally generated via a complex set of thermally induced reactions, such as the Maillard reaction, lipid oxidation, and their interaction [5]. With the increasing demand for mutton products, information regarding the flavor formation and characteristics of mutton products has attracted extensive attention. Therefore, for the sake of meeting the growing demand for meat, an increasing number of studies have focused on the flavor, sensory, and odor characteristics of meat [6]. Notably, consumers are positively affected by local mutton production and animal welfare information [7], indicating that if the advantages of consuming meat from local sheep breeds are properly communicated to consumers, the potential of the corresponding local sheep breed market will increase.

GC-IMS has demonstrated its effectiveness in the characterization and analysis of volatile compounds in different substrates, including food samples [8]. Furthermore, it might offer a viable alternative to the most common flavor analysis methods like chromatography olfactometry–mass spectrometry (GC-O-MS), gas chromatography–mass spectrometry (GC-MS), and an electronic nose (E-nose) [9]. GC-MS is the preferred volatile compound analysis technique, and it has a wide range of applications. However, due to the complexity of food matrices, complex pretreatment is usually required before analysis, and GC-MS may not meet the requirements of rapid detection [10]. Aromatic active compounds from complex mixtures can be effectively selected using GC-O-MS, but a large amount of repetitive work is needed, which results in time being wasted. Therefore, it is not appropriate for use in the detection of volatile compounds in food [11]. E-noses provide a low-cost, high sensitivity method that needs no preprocessing. Nevertheless, they still have a lot of drawbacks, including the use of membrane materials, immature manufacturing processes, and data handling programs [12]. GC-IMS has the advantages of high sensitivity, a fast analysis speed without complex preparation procedures, and a low cost compared with other analytical techniques. In addition, GC-IMS displays the results of the analysis in a color contour image, allowing for the intuitive display of the differences between samples [13]. GC-IMS has been widely applied in the study of volatile compounds in foods such as edible vegetable oils [14], Tricholoma matsutake Sing [15], rice [16], eggs [17], peppers [18], Iberian ham [19], etc.

Chemometrics models, such as OPLS-DA, are typically used for fast, simple, and efficient sample classification. The key advantage of this technique is its ability to identify tiny differences in similar samples [20]. Yang [21] showed that combining GC-IMS with OPLS-DA can rapidly discriminate between flavored fishcakes prepared using different cooking methods. The regional classification of Schizonepetae Spica using OPLS-DA shows that the GC-IMS method can classify samples better than the GC-MS method [22]. In addition, Arroyo-Manzanares [23] reported that the OPLS-DA model based on GC-IMS differentiated between pure and adulterated honey with a validation success of 97.4%. In recent years, many studies have reported on the flavor compounds in mutton. However, little has been reported about the identification, fingerprinting, and differential analysis of the volatile components in meat from different breeds of sheep according to GC-IMS; in particular, there is limited information regarding combining chemometrics techniques to distinguish between different sheep breeds.

The main objective of this research was to reveal the differences in volatile compounds in sheep meat from different breeds. Specifically, GC-IMS was used to identify the volatile compounds of different sheep breeds and visualize their volatile component fingerprints. Simultaneously, the differences between volatile components were screened through the use of OPLS-DA, which can realize the recognition of sheep breeds and provide a foundation for further study.

## 2. Materials and Methods

### 2.1. Ethical Statement

The experimental animal protocol was reviewed and approved by the Animal Administration and Ethics Committee of the Lanzhou Institute of Husbandry and Pharmaceutical Science of the Chinese Academy of Agricultural Science (Permit No. SYXK-2019-010).

### 2.2. Materials

All sheep were purchased from Jingyuan Jiumuyuan Ecological Development Co., Ltd. (Jingyuan, China). A total of 24 male sheep were selected from four breeds, including Tibetan sheep, Hu sheep, Ujumqin sheep, and Tan sheep. Lambs inherited the lineage from the same sire within the breed and were reared in the same environment. All lambs were slaughtered and weighed (24.14 ± 0.60 kg for Tibetan sheep; 29.02 ± 0.85 kg for Hu sheep; 27.24 ± 0.60 kg for Ujumqin sheep; and 26.98 ± 1.58 kg for Tan sheep), and the longissimus dorsi of each lamb (100 g) was selected as the experimental sample, which was kept frozen at −80 °C until analysis.

### 2.3. GC-IMS Analysis

A GC-IMS (Flavorspec^®^, G.A.S. Instrument, Dortmund, Germany) with an FS-SE-54-CB capillary column (15 m × 0.53 mm) was used for the volatile compounds analysis of the lamb samples. Before the experiment, all of the samples were thawed in the refrigerator at 4 °C.

Before GC-IMS analysis, the following headspace parameters were optimized, including the incubation temperature, incubation time, and headspace injection volume. Each meat sample (3.0 g) was precisely weighed and added to a 20 mL headspace vial. Next, the samples were incubated at 70 °C for 10 min. Then, 500 μL of the sample was injected with a syringe pre-heated to 80 °C. The capillary column temperature was maintained at 60 °C while the drift tube was at 45 °C. The drift gas flow rate in a drift tube was 150 mL/min. Nitrogen with 99.999% purity was used as carrier gas with the following flow rates: 0–2 min (2 mL/min), 2–10 min (5–20 mL/min), and 10–25 min (20–100 mL/min). The data were finally transferred to a computer for further analysis.

### 2.4. Statistical Analysis

The GC-IMS data were viewed and processed using Laboratory Analytical Viewer (LAV) from G.A.S. Data analysis was performed using SPSS 18.0 version (SPSS Inc., Chicago, IL, USA), and the differences among samples were analyzed via one-way ANOVA (*p* < 0.05). OPLS-DA analysis was conducted using SIMCA 14.1. Origin software (version 2019b, Microcal Inc., Amherst, MA, USA) was applied to visualize cluster analysis and heatmaps.

## 3. Results

### 3.1. GC-IMS Maps of Meats of Different Breeds of Sheep

A topographic map was used to characterize the substances in the meats of different breeds of sheep. The volatile compounds present in the meat of different breeds of sheep were obtained via GC-IMS analysis, and the data were generated in the form of a two-dimensional (2D) spectrum, as shown in Figure 1. Figure 1 represents all of the volatile compounds found in the sheep meat. The red vertical line at horizontal coordinate 1.0 is the reactive ion peak (RIP). The differences between the sheep meats could be analyzed more clearly by using the difference comparison model. The spectrogram of Tibetan sheep meat was selected for reference, and the spectrogram of the other three breeds of sheep meat was derived. The white background indicates that the two volatile compounds were identical. Red spots indicate a higher content of a substance than in the reference, while blue spots indicate a lower content of a substance. Most of the detection data appear in the retention time of 100 to 700 s and the drift time of 1.0 to 2.0 s. When the retention time was between 100 and 400 s, more red spots appeared in the topographic maps of Tan sheep and Ujumqin sheep meat. 

### 3.2. Volatile Compound Identification and Fingerprints Profile in Different Breeds’ Meat

The compounds were characterized using GC × IMS. Some single compounds might produce multiple signals (dimers or even trimers) [24]. A total of 71 peaks and 47 typical target compounds were identified from the GC × IMS library of different breeds’ meat, including 26 aldehydes, 15 ketones, 18 alcohols, eight esters, two thiazoles, one furan, and one ether (Table 1). Aldehydes, ketones, and alcohols were regarded as the major volatile compounds. All of the compounds identified in the meats of different breeds of sheep, including the name, retention index, retention time, drift times, intensity, and *p*-values, are presented in Table 1.

The fingerprint spectra showed a detailed analysis of all the signal peaks (Figure 2). In the fingerprint, the rows indicate samples, and the columns indicate volatile compounds. The content of volatile compounds is indicated by the brightness of the color, and a brighter color represents a higher content. The fingerprint can help to clarify the details of each volatile compound. The contents of 2-pentanone, 3-pentanone, 4-methyl-2-pentanone, 2-acetylthiazole, 2,3-butanediol, and 1-propene-3-methylthio were relatively elevated in the meat of Tibetan sheep compared to those found in the other three types of sheep meat. The highest levels of ethyl acetate, ethyl 2-hydroxypropanoate, ethanol, 1-hexanol, heptanol, 3-methyl-3-buten-1-ol, and butanal were found in the meat of Ujumqin sheep. The levels of 2-propanol, 1-pentanol, (E)-2-hexen-1-ol, 1-octen-3-ol, 2-methyl-1-propanol, pentanal, (E)-2-pentenal, (E)-2-heptenal, heptanal, (E)-2-octenal, octanal, n-nonanal, phenylacetaldehyde, 1-octen-3-one, 2-heptanone, 2-pentylfuran, and trimethylthiazole were abundant in the meat of Tan sheep. Furthermore, some volatile compounds, including methyl benzoate, 2-propanone, benzaldehyde, 3-methylbutanal, 2-methylbutanal, and methional, had high contents in the meat of Tibetan sheep and Hu sheep. The characteristic volatile compounds in the meat of Tan sheep and Ujumqin sheep were the highest in all types of sheep meat. The higher the content and variety of volatile compounds, the greater the flavor [25].

### 3.3. Analysis of Volatile Compounds in Sheep Meat of Different Breeds

Previous studies have shown that aldehydes are the most important substances found in the meat of ruminants [26]. Generally, the odor threshold of aldehydes is known to be relatively low; thus, they have a critical effect on the volatile flavor of mutton. Aldehydes are mainly derived from the oxidative degradation of fatty acids and can also be synthesized via the Strecker degradation reaction [27]. Strecker degradation involves the formation of aldehydes after the decarboxylation and deamination of amino acids. Only a tiny amount of phenylacetaldehyde was detected in the Tan lamb, which was synthesized via the Strecker degradation of phenylalanine. It is characterized by a sweet taste, like honey, with a low odor threshold [28]. However, the almond- and caramel-like smell of benzaldehyde might have a negative impact on the odor profile. Benzaldehyde was detected in all sheep meat, and the content was higher in the meat of Tibetan sheep and Hu sheep. There were several aldehydes related to the volatile flavor of meat, including hexanal, heptanal, octanal, and nonanal, which were mainly derived from the oxidation of fatty acids, such as oleic acid, linoleic acid, linolenic acid, and arachidonic acid [29]. Aldehydes with small molecular weights, such as 2-methylbutyraldehyde, 2-methylpropanal, and 3-methylbutyraldehyd, could be transformed from isoleucine, valine, leucine, and others [30]. Aldehydes usually possess a strong aroma; low-molecular-weight aldehydes (3-4 C) are known to have a strong, pungent flavor. For medium-molecular-weight aldehydes (5-9 C), hexanal has a light green grass and grass fragrance, octanal has a fruity flavor, and nonanal is fragrant. However, higher-molecular-weight (10-12 C) aldehydes are known to have the flavor of orange peel [31]. The content of octanal in the meat of Tan sheep and Ujmqin sheep was much higher than in the meat of Tibetan sheep, while the content of hexanal and heptanal in the meat of Hu sheep was much higher compared to that in the other three meats of different breeds of sheep. The lack of volatile compounds in the meat of Hu sheep and Tibetan sheep, especially aldehydes, was one of the key factors that caused variable odor characteristics.

Most alcohols are obtained by degrading conjugated linoleic acid in muscle through lipoxygenase and peroxidase [32]. Alcohols also account for a high proportion of volatile flavor compounds in lamb. Alcohol compounds are known to have a relatively higher odor threshold. Compared with aldehydes, their contribution to volatile flavor is weaker [33]. Still, alcohols play an essential role in influencing the olfactory perception of human odor profiles [34]. Alcohols generally have plant, rancid and chemical flavors. The flavor of straight-chain low-grade alcohols is known to increase with increasing carbon chain length, showing the characteristics of light, woody, and fatty aromas [35]. Here, 1-octene-3-ol was found in the meat of all sheep, with the lowest content found in the meat of Tibetan sheep. 1-octen-3-ol is an important alcohol-flavored substance with a mushroom-like aroma. The elevated content of alcohols or unsaturated alcohols would affect the flavor of lamb, while other alcohols would not significantly impact the flavor of lamb [27].

Ketones, as lipid oxidation products, usually have fruity, buttery, and creamy flavors. The threshold of ketones is much higher than that of their isomeric aldehydes, which have positive impacts on the flavor of meat [27,36]. The content of ketones in the sheep meat of different breeds was significantly different, having a vital impact on the flavor of sheep meat. 3-hydroxy-2-butanone is known to give sheep meat a buttery flavor, which was detected in all sheep. The content of 6-methyl-5-heptene-2-one was highest in Hu sheep but was not found in the meat of Tibetan sheep.

Ester compounds originate from the esterification reaction of carboxylic acids and alcohols or are generated during the oxidative decomposition of fat [37]. They are a common volatile component relating to the flavor of foods. Due to the fact that there are different fatty acids in meat, esters produced by long-chain fatty acids have a slightly oily flavor, while those from short-chain fatty acids, especially those with methyl-branched chains, often have a fruity flavor [38]. For example, ethyl acetate has a sweet, fruity flavor. The ethyl acetate content was found to be highest in Ujumqin lamb, very low in Tibetan sheep and Tan sheep, and was not detected in the meat of Hu sheep.

2-pentylfuran, found in Ujumqin and Tan lambs, is synthesized via the oxidative decomposition of linoleic acid as a flavor precursor. 2-pentylfuran was mainly synthesized via lipid oxidation [39]. However, in the Maillard reaction, the dehydrogenation product produced by the intermediate Amadori rearrangement product could also generate 2-pentylfuran following the addition of ammonia and hydrogen sulfide [40]. Usually, ether compounds have a pleasant aroma, and sulfur compounds enrich the flavor of meat due to their unique chemical elements [41]. When meat is heated, thiazole, thiophene, furan, and sulfur compounds are formed due to the degradation and rearrangement of the amino acids as precursors. Ribose and cysteine, as precursors, can produce sulfide, becoming critical flavor compounds in meat products [42]. Only one ether compound, 1-propene-3-methylthio, was found in all sheep, and the levels of 1-propene-3-methylthio were highest in the meat of Tibetan sheep. The content of trimethylthiazole was highest in the meat of Tan sheep, and only 2-acetylthiazole was found in the meat of Ujumqin sheep.

### 3.4. Principal Component Analysis (PCA) of Volatile Compounds

Fingerprints can be used to distinguish between the meats of different breeds of sheep and determine the differences in their volatile compound content. However, the graphical expression of 2-D GC-IMS data (Figure 1) is influenced by the subjective cognition of observers. The FlavourSpec^®^ PCA classification model built into the food flavor analyzer was used to determine the qualitative and non-qualitative volatile compounds of sheep meat, respectively (Figure 3). The total variance in the contribution of the first two principal components was 86%, indicating that the PCA results were effective.

From the degree of the aggregation and dispersion of the samples, the meats of different breeds of sheep were distributed in different quadrants, with obvious separation, indicating that there were some differences in volatile compounds among the meats of different breeds of sheep. Previous studies have shown that the VOCs in meat are breed-dependent [43]. Thus, selected compounds in sheep meat could be used as biomarkers to distinguish different sheep breeds.

### 3.5. OPLS-DA of Volatile Compounds

OPLS-DA is considered to be an effective method for sample classification and the establishment of discriminant models, which visualize the degree of difference between samples based on the correlation between data [44]. R^2^X and R^2^Y represent the explanation percentages of the fitting equations for X and Y matrices, respectively, while Q^2^ indicates the predictive ability of the fitting equation [45]. The results were deemed accurate when the coefficients of R^2^ and Q^2^ exceeded 0.5 and approached 1.0. As shown in Figure 4A, R^2^Y = 0.988, Q^2^ = 0.974 and are both close to 1, indicating that the model has good predictive ability and explanatory ability. The sheep meat of different breeds could be well separated. Ujumqin and Tan were located in the second and third quadrants, respectively, and Hu and Tibetan were located in the fourth quadrant. To avoid overfitting, the reliability of OPLS-DA was tested by performing 200 cross-substitution tests on the model (Figure 4B). The intercept between the regression line of model Q^2^ and the horizontal coordinate is negative (−0.426), and all the substitution tests’ R^2^ and Q^2^ are lower than the original values, indicating that the model is stable and reliable and without overfitting [21].

### 3.6. Screening of Differential Volatile Components in Different Breeds of Sheep Meat

After analyzing the 71 volatile components found in the meats of different breeds of sheep through the use of GC-IMS and the construction of a reliable OPLS-DA simulation, the impact of each component on classification was quantified according to VIP. The VIP value is used to distinguish and classify the volatile components of the samples. The higher the VIP value, the greater the difference between different volatile components [46]. Figure 5A shows the results for the VIP values of the key components. 3-hydroxy-2-butanone (dimer), ethyl acetate (dimer), hexanal (dimer), octanal (dimer), ethanol, heptanal (dimer), 2-butanone (dimer), n-nonanal (both monomer and dimer), pentanal (dimer), ethanol (dimer), octanal, 3-methylbutanal (dimer), 2-butanone, pentanal, 1-propene-3-methylthio, 1-hexanol (dimer), hexanol, heptanal, 2-propanonea, and methyl-2-butan-1-ol have VIP values greater than 1. In addition, PCA and heatmap clustering analyses were conducted based on the 21 different volatile compounds. Most of the differences in the samples can be discriminated using PCA, with a total ratio of 76% (PC1 and PC2 were 44.7% and 31.3%, respectively) (Figure 5B). The heatmap clustering results also indicated that the 21 kinds of labeled volatile components in the meats of different breeds of sheep could better classify the differences in the samples (Figure 5C). Therefore, combining PCA and cluster analysis with these labeled volatile components can effectively distinguish the differences between different breeds of sheep meat.

## 4. Conclusions

In conclusion, this experiment identified differences in volatile components of different breeds of sheep meat by GC-IMS technique. A total of 71 volatile organic chemicals were detected in the meats of different breeds of sheep, including 26 aldehydes, 18 alcohols, 15 ketones, eight esters, one ether, one furan, and two thiazoles. The variations in volatile components in the meats of different breeds of sheep might be well recognized via the use of GC-IMS data together with PCA. GC-IMS screened 21 compounds (VIP > 1) as potential signature compounds, which could well reflect the differences in volatile components in the meats of different breeds of sheep. The results of this study contribute to a better understanding of the volatile compound characteristics of sheep meat and provide valuable information for improving the flavor of sheep meat. In addition, combining GC-IMS data with chemometrics has important potential in volatile compound-based sheep breed identification. This was a preliminary study to help flavor researchers and consumers understand the aroma profiles of sheep meat and the aromatic active compounds that need to be identified. Furthermore, in subsequent research, it is necessary to establish a detailed volatile compound database of different sheep breeds based on GC-MS, GC-IMS, GC-O-MS, electronic tongues, and electronic noses and elucidate the formation mechanism of aroma substances through the use of multi-omics methods.

## Figures and Tables

**Figure 1 foods-13-02647-f001:**
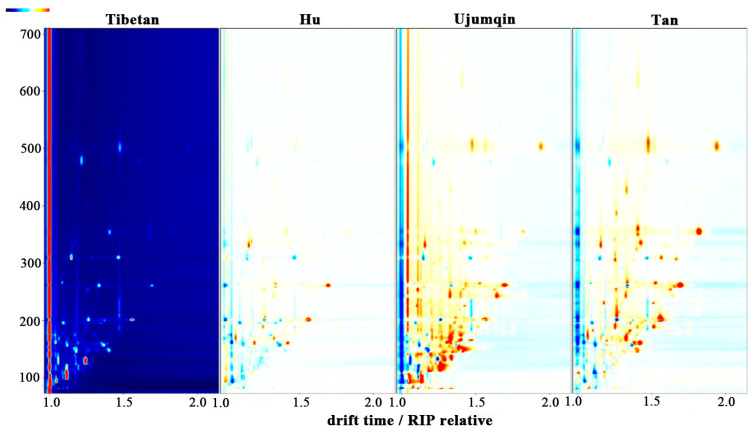
Two-dimensional topographic representation of different breeds’ meat.

**Figure 2 foods-13-02647-f002:**
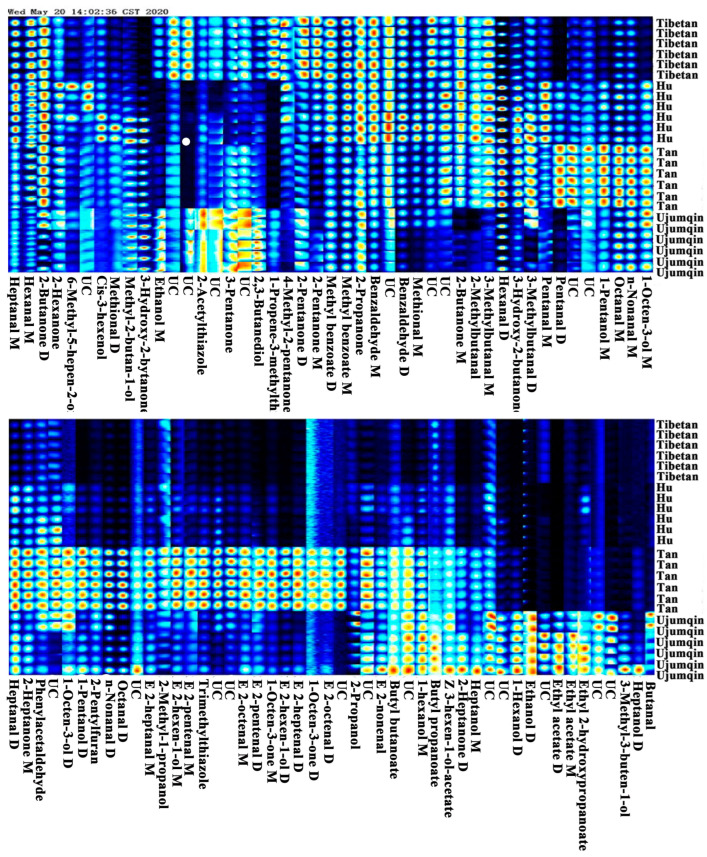
Fingerprint of volatile compounds in the meat of different breeds of sheep (UC: unidentified compound).

**Figure 3 foods-13-02647-f003:**
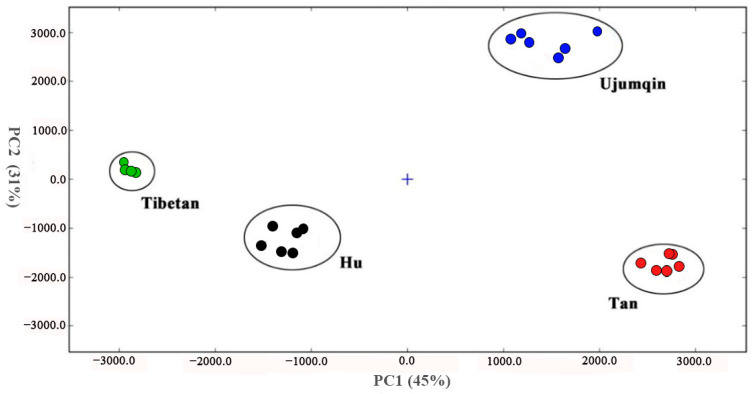
PCA plot of volatile compounds in the meat of different breeds of sheep.

**Figure 4 foods-13-02647-f004:**
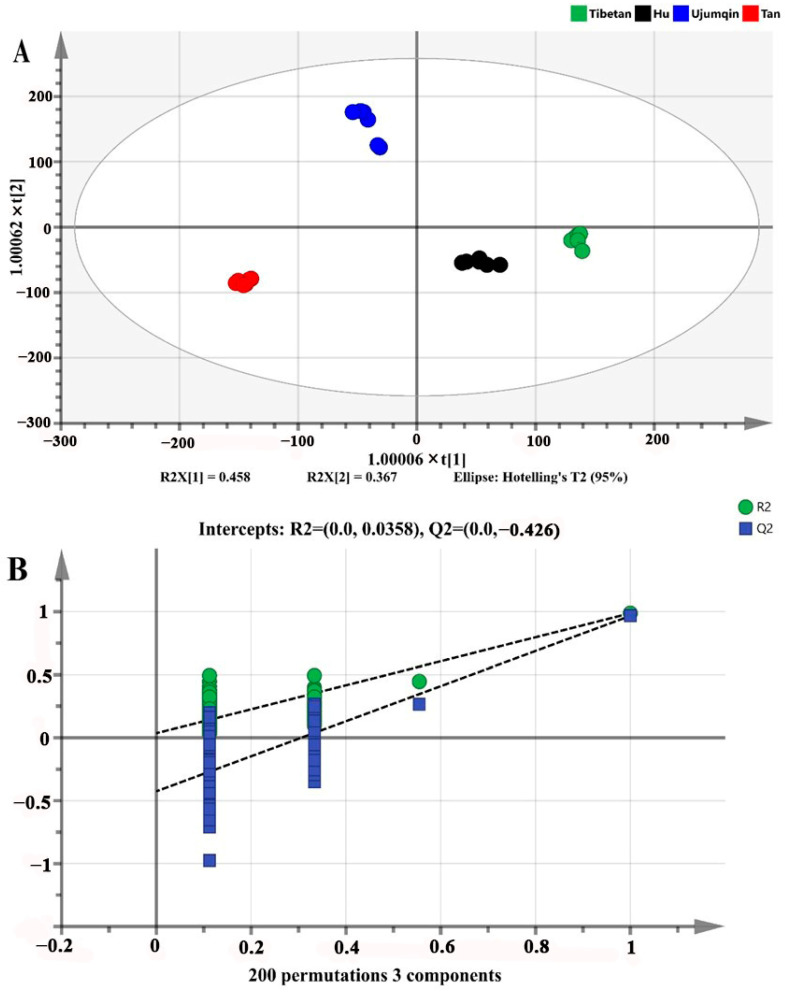
(**A**) Score plot of OPLS-DA (R^2^Y = 0.9888; Q2 = 0.974); (**B**) cross-substitution plot of 200 permutation tests (R^2^ = 0.00358; Q^2^ = 0.426).

**Figure 5 foods-13-02647-f005:**
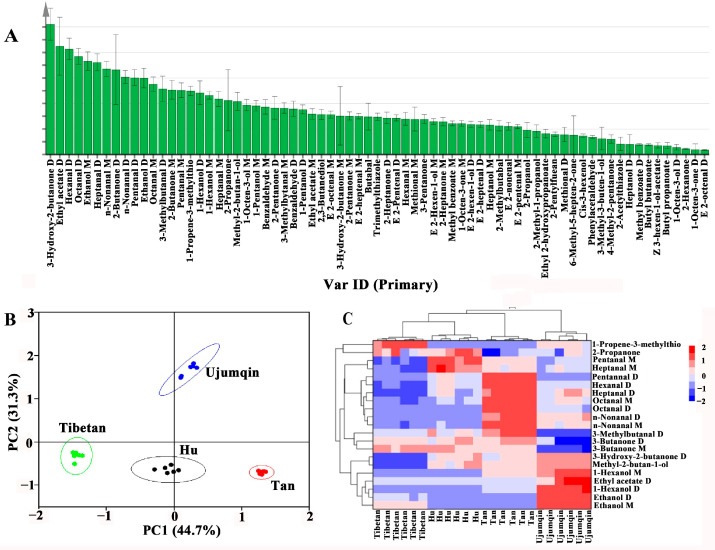
Screening of differential volatile components in different breeds of sheep meat (**A**) VIP value. (**B**) PCA score plot. (**C**) Clustering heatmap.

**Table 1 foods-13-02647-t001:** List of volatile compounds identified via GC-IMS in the meat of different breeds of sheep (*n* = 6).

No.	Compounds	Retention Index Was	Retention Times (s)	Drift Times (ms)	Intensity (V)				*p*-Value
					**Tibetan**	**Hu**	**Ujumqin**	**Tan**	
Aldehydes	Aldehydes								
1	n-Nonanal(monomer)	1111.6	503.873	1.4763	852.90 ± 157.8 ^d^	1228.2 ± 114.7 ^c^	2176.0 ± 211.5 ^b^	3484.9 ± 111.6 ^a^	<0.001
2	n-Nonanal(dimer)	1111.6	503.873	1.9430	–	–	1009.5 ± 122.0	1804.3 ± 158.5	–
3	Octanal(monomer)	1001.5	356.663	1.4052	393.26 ± 72.23 ^d^	871.25 ± 119.5 ^c^	1158.2 ± 222.6 ^b^	2345.9 ± 28.70 ^a^	<0.001
4	Octanal(dimer)	1000.3	355.394	1.8268	–	161.20 ± 30.43	801.52 ± 349.6	3087.3 ± 150.0	–
5	Heptanal(monomer)	896.6	262.930	1.3380	714.59 ± 69.54 ^d^	1194.8 ± 88.76 ^a^	890.21 ± 59.43 ^c^	980.08 ± 20.32 ^b^	<0.001
6	Heptanal(dimer)	895.0	261.717	1.6961	274.53 ± 71.29 ^d^	1507.0 ± 431.4 ^c^	2325.0 ± 454.3 ^b^	3630.0 ± 79.62 ^a^	<0.001
7	3-Methylbutanal(monomer)	657.4	147.779	1.1822	1071.1 ± 25.77 ^c^	1169.2 ± 30.82 ^a^	448.21 ± 48.05 ^d^	1115.9 ± 19.20 ^b^	<0.001
8	3-Methylbutanal(dimer)	659.1	148.294	1.4031	745.22 ± 47.10 ^c^	1225.4 ± 269.2 ^b^	208.06 ± 21.69 ^d^	1883.3 ± 81.89 ^a^	<0.001
9	Hexanal(monomer)	796.7	202.150	1.2638	874.95 ± 22.03 ^b^	952.69 ± 33.85 ^a^	586.19 ± 30.34 ^d^	663.80 ± 18.41 ^c^	<0.001
10	Hexanal(dimer)	797.4	202.513	1.5595	1137.6 ± 161.8 ^d^	3018.3 ± 254.0 ^b^	2592.9 ± 156.3 ^c^	5392.6 ± 107.6 ^a^	<0.001
11	Pentanal(monomer)	699.5	161.350	1.1953	257.04 ± 27.81 ^d^	884.52 ± 66.76 ^a^	433.57 ± 72.08 ^c^	594.56 ± 13.24 ^b^	<0.001
12	Pentanal(dimer)	699.5	161.350	1.4224	–	698.19 ± 123.9	138.75 ± 18.14	2091.4 ± 26.58	–
13	2-Methylbutanal	671.4	152.074	1.1669	297.12 ± 7.320 ^b^	378.64 ± 56.69 ^a^	93.190 ± 38.57 ^c^	306.62 ± 12.66 ^b^	<0.001
14	(E)-2-Octenal(monomer)	1060.3	427.724	1.3303	–	–	60.160 ± 21.04	466.27 ± 29.40	–
15	(E)-2-Octenal(dimer)	1060.6	428.108	1.8179	–	–	–	5.7300 ± 0.3700	–
16	(E)-2-Nonenal	1176.0	619.851	1.4068	–	–	199.44 ± 43.18	224.92 ± 48.26	–
17	(E)-2-Heptenal(monomer)	951.5	307.387	1.2535	–	44.600 ± 11.79	185.27 ± 65.85	501.61 ± 14.21	–
18	(E)-2-Heptenal(dimer)	952.1	307.879	1.6690	–	–	16.530 ± 9.510	255.94 ± 19.08	–
19	(E)-2-Pentenal(monomer)	755.4	182.988	1.1027	20.150 ± 4.210 ^c^	34.840 ± 10.47 ^c^	128.13 ± 21.77 ^b^	273.50 ± 8.970 ^a^	<0.001
20	(E)-2-Pentenal(dimer)	755.2	182.891	1.3552	–	–	164.65 ± 27.32	401.62 ± 24.71	–
21	Methional(monomer)	902.2	267.016	1.0886	249.87 ± 40.72 ^b^	402.23 ± 88.24 ^c^	41.190 ± 14.93 ^a^	219.02 ± 10.17 ^b^	<0.001
22	Methional(dimer)	901.1	266.212	1.3970	–	82.540 ± 33.22	79.580 ± 16.08	91.040 ± 6.470	–
23	Benzaldehyde(monomer)	955.5	310.997	1.1478	1338.8 ± 54.49 ^a^	1308.5 ± 110.7 ^a^	630.29 ± 140.7 ^c^	995.33 ± 20.19 ^b^	<0.001
24	Benzaldehyde(dimer)	956.5	311.847	1.4698	861.81 ± 110.0 ^b^	995.85 ± 114.7 ^a^	241.36 ± 106.0 ^d^	638.56 ± 28.82 ^c^	<0.001
25	Phenylacetaldehyde	1042.8	404.911	1.2571	–	–	–	85.830 ± 1.570	–
26	Butanal	632.5	140.723	1.1124	46.070 ± 12.49 ^b^	51.040 ± 10.97 ^b^	541.50 ± 139.6 ^a^	21.980 ± 8.920 ^b^	<0.001
	Alcohols								
27	1-Octen-3-ol(monomer)	978.2	332.525	1.1579	23.380 ± 3.270 ^d^	527.47 ± 92.48 ^c^	721.45 ± 172.5 ^b^	875.78 ± 29.51 ^a^	<0.001
28	1-Octen-3-ol(dimer)	977.4	331.738	1.5990	–	–	–	15.280 ± 5.180	–
29	Ethanol(monomer)	355.2	94.087	1.0473	1309.1 ± 20.07 ^b^	178.19 ± 64.75 ^d^	2799.2 ± 116.4 ^a^	509.92 ± 76.29 ^c^	<0.001
30	Ethanol(dimer)	353.9	93.969	1.1330	57.740 ± 8.370	20.100 ± 5.530	1853.8 ± 57.72	–	–
31	1-Hexanol(monomer)	870.0	244.461	1.3263	–	121.97 ± 53.86	1203.0 ± 95.83	744.45 ± 16.20	–
32	1-Hexanol(dimer)	869.3	243.958	1.6447	–	–	1195.3 ± 193.5	66.860 ± 7.360	–
33	Methyl-2-butan-1-ol	722.3	169.615	1.2336	–	472.66 ± 72.18	578.06 ± 24.30	414.11 ± 23.15	–
34	Heptanol(monomer)	970.7	325.203	1.4007	–	–	265.75 ± 73.50	208.75 ± 10.02	–
35	Heptanol(dimer)	970.3	324.829	1.7668	–	–	39.870 ± 38.22	–	–
36	(E)-2-Hexen-1-ol(monomer)	849.0	231.146	1.1791	67.900 ± 5.440 ^d^	108.81 ± 22.80 ^c^	173.72 ± 17.84 ^b^	436.25 ± 13.92 ^a^	<0.001
37	(E)-2-Hexen-1-ol(dimer)	849.0	231.146	1.5160	–	–	–	253.87 ± 21.38	–
38	1-Pentanol(monomer)	769.4	189.159	1.2548	63.480 ± 21.04 ^d^	434.02 ± 57.26 ^c^	486.54 ± 50.54 ^b^	10008 ± 11.86 ^a^	<0.001
39	1-Pentanol(dimer)	769.5	189.191	1.5160	–	21.750 ± 6.060	267.17 ± 146.3	660.26 ± 32.92	–
40	*Cis*-3-hexenol	866.2	241.985	1.2291	–	44.930 ± 2.260	–	–	–
41	2-Methyl-1-propanol	629.0	139.767	1.1732	200.34 ± 32.43 ^b^	223.72 ± 48.44 ^b^	176.87 ± 62.28 ^b^	372.14 ± 32.26 ^a^	<0.001
42	2-Propanol	486.8	110.348	1.1797	148.67 ± 13.74 ^b^	191.69 ± 49.32 ^b^	361.86 ± 120.9 ^a^	380.49 ± 35.08 ^a^	<0.001
43	3-Methyl-3-buten-1-ol	739.4	176.297	1.5005	–	–	86.540 ± 74.51	–	–
44	2,3-Butanediol	796.1	201.865	1.3666	190.41 ± 36.31 ^c^	147.13 ± 66.19 ^c^	695.25 ± 52.24 ^a^	253.54 ± 7.260 ^b^	<0.001
	Ketones								
45	2-Pentanone(monomer)	687.6	157.292	1.1198	635.89 ± 13.82 ^a^	339.10 ± 27.00 ^b^	275.64 ± 104.3 ^b^	73.940 ± 5.340 ^c^	<0.001
46	2-Pentanone(dimer)	689.5	157.941	1.3672	803.69 ± 58.28 ^a^	394.90 ± 49.01 ^b^	709.80 ± 110.2 ^a^	430.08 ± 14.15 ^b^	<0.001
47	3-Pentanone	700.2	161.577	1.3504	270.94 ± 25.81 ^c^	133.72 ± 11.57 ^d^	476.41 ± 42.06 ^a^	348.99 ± 12.20 ^b^	<0.001
48	2-Heptanone(monomer)	885.6	255.072	1.2622	59.490 ± 10.90 ^d^	210.50 ± 25.50 ^c^	252.10 ± 49.42 ^b^	496.72 ± 13.35 ^a^	<0.001
49	2-Heptanone(dimer)	884.4	254.232	1.6306	–	–	427.75 ± 79.73	254.21 ± 18.43	–
50	1-Octen-3-one(monomer)	1029.4	388.481	1.2639	–	–	–	268.88 ± 9.850	–
51	1-Octen-3-one(dimer)	1029.1	388.096	1.6851	–	–	–	9.4600 ± 1.010	–
52	3-Hydroxy-2-butanone(monomer)	721.9	169.481	1.0655	1823.5 ± 857.3 ^a^	2115.7 ± 541.0 ^a^	847.44 ± 79.35 ^a^	1968.6 ± 38.99 ^a^	0.478
53	3-Hydroxy-2-butanone(dimer)	722.3	169.626	1.3298	–	2787.35 ± 377.1	3898.6 ± 122.5	2541.7 ± 127.3	–
54	2-Hexanone	787.2	197.492	1.1909	102.41 ± 4.460 ^b^	108.82 ± 16.81 ^ab^	127.09 ± 28.21 ^a^	115.72 ± 7.060 ^ab^	0.105
55	2-Butanone(monomer)	591.0	130.333	1.0590	1666.1 ± 36.94 ^a^	1684.9 ± 86.22 ^a^	271.69 ± 45.84 ^c^	1381.3 ± 32.19 ^b^	<0.001
56	2-Butanone(dimer)	587.2	129.463	1.2448	4543.0 ± 290.1 ^a^	4637.8 ± 428.7 ^a^	2513.1 ± 262.0 ^b^	5002.0 ± 167.5 ^a^	<0.001
57	2-Propanone	481.7	109.558	1.1164	3151.0 ± 397.2 ^ab^	3378.9 ± 269.9 ^a^	2849.6 ± 271.0 ^bc^	2430.6 ± 502.7 ^c^	0.002
58	4-Methyl-2-pentanone	740.3	176.683	1.1776	171.89 ± 14.80 ^a^	151.58 ± 23.41 ^a^	84.480 ± 13.38 ^b^	101.50 ± 11.93 ^b^	<0.001
59	6-Methyl-5-hepten-2-one	984.7	339.068	1.1773	–	104.24 ± 29.93	50.100 ± 27.64	37.020 ± 13.96	–
	Esters								
60	Ethyl acetate(monomer)	620.2	137.471	1.0946	11.730 ± 2.600	–	506.50 ± 111.6	6.3400 ± 2.500	–
61	Ethyl acetate (dimer)	614.2	135.942	1.3345	–	–	3536.6 ± 324.6	–	–
62	ethyl 2-hydroxypropanoate	810.9	209.472	1.1442	–	–	134.89 ± 33.77	3.9900 ± 1.030	–
63	Butyl propanoate	873.9	247.025	1.2849	–	–	24.740 ± 12.12	–	–
64	(Z)3-Hexen-1-ol-acetate	984.3	338.625	1.3063	–	–	26.980 ± 7.070	3.3300 ± 1.620	–
65	Methyl benzoate(monomer)	1097.2	481.030	1.2179	1005.26 ± 20.60 ^a^	936.50 ± 19.08 ^b^	690.08 ± 22.35 ^c^	709.72 ± 15.68 ^c^	<0.001
66	Methyl benzoate(dimer)	1096.8	480.396	1.6071	38.160 ± 4.010	26.630 ± 3.900	10.650 ± 2.700	–	–
67	Butyl butanoate	988.6	343.068	1.3363	–	–	18.420 ± 4.620	28.180 ± 170.0	–
	Furan								
68	2-Pentylfuran	988.5	342.924	1.2520	–	–	48.520 ± 5.390	123.94 ± 9.370	–
	Ether								
69	1-Propene-3-methylthio	701.6	162.080	1.0425	807.79 ± 26.87 ^a^	37.280 ± 11.45 ^c^	480.16 ± 78.12 ^b^	21.380 ± 1.260 ^c^	<0.001
	Thiazole								
70	Trimethylthiazole	955.3	310.770	1.5658	–	35.100 ± 7.700	80.390 ± 20.60	455.97 ± 21.06	–
71	2-Acetylthiazole	1018.8	375.938	1.1294	–	–	45.160 ± 22.38	–	–

^a,b,c^ Means with different letters within a row differ significantly (*p* < 0.05); **–** Represents undetected compound.

## Data Availability

The original contributions presented in the study are included in the article, further inquiries can be directed to the corresponding authors.

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
