# Peer review of "Characterization of Flavor Compounds in Chinese Indigenous Sheep Breeds Using Gas Chromatography–Ion Mobility Spectrometry and Chemometrics"

_foods, 2024, doi:10.3390/foods13172647_

Round 1
Reviewer 1 Report
Comments and Suggestions for Authors
In the present manuscript, the authors present the potential of using chromatography-ion migration spectrometry and chemometrics in the analysis of flavor compounds in Chinese indigenous sheep breeds.
The authors have demonstrated the potential benefits that gas chromatography-ion migration spectrometry offers. However, there remain a few errors that require correction:
- In the title of the manuscript, Line 2, add ’’by gas chromatography-ion migration’’, instead of ’’by chromatography-ion migration’’.
- Line 146: The sentence is not the clearest; I assume that they wanted to refer the reader to the next section or the pictures in it. In that scenario, it would be more appropriate to point to a specific figure or scheme, such as Fig. 2.
- Figures 1 and 2 show the same result in two ways. I strongly recommend the authors use Figure 1 for the graphic abstract (it is certainly optional) and leave Figure 2 as a representation of the results in the text itself.
- Table 1. Values of measured intensities (V) and standard deviations in individual samples should be displayed with the first significant figure; it will be much clearer for the reader.
The general comment is that the resolution of images 1, 2 and 3 is very poor and should be replaced.
Author Response
Manuscript ID: Response to Reviewer Comments
Manuscript ID: foods-3137446
Type of manuscript: Article
Title: “Characterization of flavor compounds in Chinese indigenous sheep breeds by gas chromatography-ion migration spectrometry and chemometrics”.
Dear Editor,
Thank you give us opportunity to submit a revised manuscript title " Characterization of flavor compounds in Chinese indigenous sheep breeds by gas chromatography-ion migration spectrometry and chemometrics" to the “Foods”journal. We would like to thank you and the reviewer for the valuable and useful comments given with regards to our manuscript. We have made the following modifications and additions to our manuscript. We have also addressed the specific comments raised by the reviewer and the detailed responses are listed below and in the revised manuscript, we have marked all the changed sentences or words with red color.
The following are point-to-point response to the reviewer’s comments.
Comments 1: In the title of the manuscript, Line 2, add “by gas chromatography-ion migration’’, instead of” by chromatography-ion migration’’.
Response 1: Thank you for pointing this out. We agree with this comment. Therefore, we have added “by gas chromatography-ion migration’’ and marked in red in lines 2.
Comments 2: Line 146: The sentence is not the clearest; I assume that they wanted to refer the reader to the next section or the pictures in it. In that scenario, it would be more appropriate to point to a specific figure or scheme, such as Fig. 2.
Response 2: Thank you for pointing this out. We agree with this comment. We are very sorry for our negligence, “3.2. Figures, Tables and Schemes” is redundant and we have removed it.
Comments 3: Figures 1 and 2 show the same result in two ways. I strongly recommend the authors use Figure 1 for the graphic abstract (it is certainly optional) and leave Figure 2 as a representation of the results in the text itself.
Response 3: Thank you for pointing this out. We agree with this comment. We leaved Figure 2 as a representation of the results in the text itself.
Comments 4: Table 1. Values of measured intensities (V) and standard deviations in individual samples should be displayed with the first significant figure; it will be much clearer for the reader.
Response 4: Thank you for pointing this out. We agree with this comment. Therefore, The table 1. has been revised according to the reviewer’s comment.
Comments 5: The general comment is that the resolution of images 1, 2 and 3 is very poor and should be replaced.
Response 5: Thank you for pointing this out. We agree with this comment. Due to the images were automatically generated by the instrument, we are unable to improve their resolution. However, we have increased the font size, and hope to be helpful to reviewers, editors, and readers.
Reviewer 2 Report
Comments and Suggestions for Authors
Dear Authors,
In general, the manuscript is good to read, the structure of the work is clear. The authors performed a lot of determinations and analyses. The statistical analysis is good. The literature review is satisfactory.
1. Keywords. I suggest adding the following keywords: Principal Component Analysis
2. "The main objectives of the present research chiefly were (1) to identify volatile compounds in different breeds of sheep meat using GC–IMS; (2) to visualize their volatile component fingerprints of different sheep breeds; and (3) to discriminate between different breeds of sheep meat based on intensity of volatile compounds using OPLS-DA". - The aim of the work set in this way seems too obvious. Without performing the markings, it can be assumed in 99% that the instrumental technique used in these studies will be able to distinguish meat by origin. Such identification should have some important purpose, e.g. due to the method of feeding, due to the taste values marked in the sensory panel, due to the content of undesirable substances, etc.
3. There is also a lack of a reference method to GC-IMS. It could be, for example, an electronic nose or another instrumental technique.
4. Figure 3 and 6. The names of the compounds are written in too small a font. If possible, please increase the font size.
5. Figure 4. The main components PC1 and PC2 usually define some feature, in this case it could be e.g. origin of meat. Please comment.
6. The "Conclusion" section shows that the instrumental method used differentiates the meat. Nothing more. It is a bit too little for a scientific paper. As I wrote above, some interesting research hypothesis should be put forward in the manuscript. This is missing here.
Author Response
Manuscript ID: Response to Reviewer Comments
Manuscript ID: foods-3137446
Type of manuscript: Article
Title: “Characterization of flavor compounds in Chinese indigenous sheep breeds by gas chromatography-ion migration spectrometry and chemometrics”.
Dear Editor,
Thank you give us opportunity to submit a revised manuscript title " Characterization of flavor compounds in Chinese indigenous sheep breeds by gas chromatography-ion migration spectrometry and chemometrics" to the “Foods”journal. We would like to thank you and the reviewer for the valuable and useful comments given with regards to our manuscript. We have made the following modifications and additions to our manuscript. We have also addressed the specific comments raised by the reviewer and the detailed responses are listed below and in the revised manuscript, we have marked all the changed sentences or words with red color.
The following are point-to-point response to the reviewer’s comments
Comments 1: Keywords. I suggest adding the following keywords: Principal Component Analysis
Response 1: Thank you for pointing this out. We agree with this comment. Therefore, we have added “Principal Component Analysis’’ in keywords and marked in red in lines 33.
Comments 2: "The main objectives of the present research chiefly were (1) to identify volatile compounds in different breeds of sheep meat using GC–IMS; (2) to visualize their volatile component fingerprints of different sheep breeds; and (3) to discriminate between different breeds of sheep meat based on intensity of volatile compounds using OPLS-DA". - The aim of the work set in this way seems too obvious. Without performing the markings, it can be assumed in 99% that the instrumental technique used in these studies will be able to distinguish meat by origin. Such identification should have some important purpose, e.g. due to the method of feeding, due to the taste values marked in the sensory panel, due to the content of undesirable substances, etc.
Response 2: Thank you for pointing this out. We agree with this comment. Therefore, we have modified sentences in lines 89-93 and marked in red.
Comments 3: There is also a lack of a reference method to GC-IMS. It could be, for example, an electronic nose or another instrumental technique.
Response 3: Thank you for pointing this out. We agree with this comment. Therefore, we have supplemented reference method to GC-IMS in lines 61-69 and marked in red.
Comments 4: Figure 3 and 6. The names of the compounds are written in too small a font. If possible, please increase the font size.
Response 4: Thank you for pointing this out. We agree with this comment. Therefore, we have increased the font size according to the reviewer’s comment.
Comments 5: Figure 4. The main components PC1 and PC2 usually define some feature, in this case it could be e.g. origin of meat. Please comment.
Response 5: Thank you for pointing this out. We agree with this comment. Therefore, we have supplemented discussion in lines 246-258 and marked in red.
Comments 6: The "Conclusion" section shows that the instrumental method used differentiates the meat. Nothing more. It is a bit too little for a scientific paper. As I wrote above, some interesting research hypothesis should be put forward in the manuscript. This is missing here.
Response 6: Thank you for pointing this out. We agree with this comment. Therefore, we have supplemented research hypothesis in lines 302-316 and marked in red.
We have tried our best to improve the manuscript according to the comments of the reviewers. We hope that we have addressed the reviewers’ comments to a satisfactory level and looking forward to the acceptance and publication of our manuscript in Foods. We deeply appreciate for your time and help.
Thank you
Sincerely
Round 2
Reviewer 2 Report
Comments and Suggestions for Authors
Dear Authors,
Overall, the manuscript looks much better after corrections. In this form it is already suitable for publication. In the future, I suggest that the Authors correct the manuscripts in the form of review tracking, so that the reader can see what corrections have been made to the original text. I hope that the Authors' next studies will show new interesting discoveries in the field of analysis of volatile substances of different types of meat, due to the region of breeding, food consumed, etc.